# Active Contours Connected Component Analysis Segmentation Method of Cancerous Lesions in Unsupervised Breast Histology Images

**DOI:** 10.3390/bioengineering12060642

**Published:** 2025-06-12

**Authors:** Vincent Majanga, Ernest Mnkandla, Zenghui Wang, Donatien Koulla Moulla

**Affiliations:** Department of Computer Science, University of South Africa, Preller Street, Muckleneuk Ridge, Pretoria 1709, South Africa; majanvi@unisa.ac.za (V.M.); wangz@unisa.ac.za (Z.W.); moulldk@unisa.ac.za (D.K.M.)

**Keywords:** data augmentation, stain normalization technique, active contours segmentation, deep learning

## Abstract

Automatic segmentation of nuclei on breast cancer histology images is a basic and important step for diagnosis in a computer-aided diagnostic approach and helps pathologists discover cancer early. Nuclei segmentation remains a challenging problem due to cancer biology and the variability of tissue characteristics; thus, their detection in an image is a very tedious and time-consuming task. In this context, overlapping nuclei objects present difficulties in separating them by conventional segmentation methods; thus, active contours can be employed in image segmentation. A major limitation of the active contours method is its inability to resolve image boundaries/edges of intersecting objects and segment multiple overlapping objects as a single object. Therefore, we present a hybrid active contour (connected component + active contours) method to segment cancerous lesions in unsupervised human breast histology images. Initially, this approach prepares and pre-processes data through various augmentation methods to increase the dataset size. Then, a stain normalization technique is applied to these augmented images to isolate nuclei features from tissue structures. Secondly, morphology operation techniques, namely erosion, dilation, opening, and distance transform, are used to highlight foreground and background pixels while removing overlapping regions from the highlighted nuclei objects on the image. Consequently, the connected components method groups these highlighted pixel components with similar intensity values and assigns them to their relevant labeled component to form a binary mask. Once all binary-masked groups have been determined, a deep-learning recurrent neural network (RNN) model from the Keras architecture uses this information to automatically segment nuclei objects having cancerous lesions on the image via the active contours method. This approach, therefore, uses the capabilities of connected components analysis to solve the limitations of the active contour method. This segmentation method is evaluated on an unsupervised, augmented human breast cancer histology dataset of 15,179 images. This proposed method produced a significant evaluation result of 98.71% accuracy score.

## 1. Introduction

Over the past few years, there has been an increase in the demand for the early detection of breast cancer (BC) at screening sites/hospitals, thus opening avenues for new research. Early cancer detection increases the chances of making the right decisions for a successful treatment plan. Therefore, screening procedures can be analyzed through computer-aided (CAD) systems, which use medical images to improve clinical efficiency and confidentiality. Pathologists’ evaluation of medical images is quite time-inefficient and varies from person to person. Hence, most of the current research work has focused on the analysis of medical images, specifically to aid in clinical diagnosis [1].

The hematoxylin and eosin (H&E) staining procedure has shown significant results as the preferred standard for histologic examination of human glands/tissues [2]. Early diagnosis of BC relies heavily on how cancerous lesions spread on histology images, specifically nuclei and tissue glands, and thus offers a need for a prognosis. The increased need to classify cancer as either malignant or benign has led researchers to study the application of machine-learning methods to model the progression and treatment of cancerous ailments.

Deep-learning (DL) techniques have had increased success in various fields, namely object detection and image recognition, among others. CAD systems of detecting cancer based on histology images have several research prospects, and this study’s focus is on examining tissue characteristics, nuclei cell detection, and segmentation for the presence of cancer. Consequently, this study also focuses on identifying and separating these regions from histology images, thus resulting in early cancer diagnosis.

Segmentation of nuclei is an invaluable step towards automated analysis of BC histology images, and various approaches have been proposed. Most of these methods revolve around watershed segmentation, active contours, and pixel groupings combined with different morphology operations [3]. However, these techniques suffer from over-segmentation and thus do not work well for overlapping cells. Active contours have been increasingly used in image segmentation, their major limitation being the inability to settle the edges of intersecting objects, thus segmenting them as one single object [4].

In histopathology, the morphological appearance of different features and structures on an image, such as nuclei, cells, or glands, often shows the presence of disease. In the case of BC, shape and morphological characteristics of nuclei on histology images correlate with disease aggressiveness [5]. Conventionally, thresholding and morphology operations have been the most widely used techniques for image segmentation. Research authors in [6] propose morphology operations to pre-process images, threshold them, and further post-process these images to detect edges. Further, researchers in [7] propose an automated segmentation technique that uses gray scaling, median filtering, and bottom-top hat filters as pre-processing steps to enhance image contrast. Thresholding is used to identify regions of interest, while post-processing morphology techniques, namely dilation, area opening, and filling in holes, are used to improve final segmentation results.

Authors in [8] also propose thresholding and binary morphological operations, namely, dilation and erosion, to identify the breast region of interest (ROI), masking and isolating it from unwanted pectoral muscle regions. In [9], image pre-processing is handled through normalization, segmentation through color deconvolution for nuclei enhancement, and data augmentation to increase dataset size, and a binary threshold is used to detect nuclei edges in the images.

Hence, CAD systems are important for predicting BC by accurately and efficiently detecting and identifying locations of nuclei objects and segmenting them so that relevant morphological features related to BC may be obtained.

Given the importance and several challenges of segmenting cancerous nuclei in breast histology images, this paper proposes an improved active contour (connected component + active contours) method to segment cancerous lesions in unsupervised human breast histology images.

This proposed method pre-processes images through various augmentation methods, namely rotation, shifting, and scaling, to increase the size of the dataset. The H&E stain normalization technique is applied to these augmented images to separate and isolate nuclei features from tissue structures. Morphology operations, namely erosion, dilation, opening, and distance transform, are then used to highlight foreground and background pixels on the image.

Consequently, the connected components method is introduced to group highlighted pixel components with similar intensity values from the previous morphology-operated step and assigns them to their relevant labeled component (binary mask). The connected component removes overlapping regions from the remaining highlighted nuclei objects. Once all remaining nuclei objects and binary masked groups (foreground, background) have been determined, the RNN model from the Keras architecture uses gathered information to automatically segment nuclei objects having cancerous lesions on the image via the active contours method. Therefore, this proposed approach uses the capabilities of connected components analysis to group objects with similar characteristics to resolve the limitation of inaccurate segmentation of overlapping objects, which is common with the active contour method.

The main contributions of this study are to resolve intersecting nuclei objects and their edges’ inhomogeneity. Hence, assisting in correctly identifying, isolating, segmenting, and detecting cancerous lesions on BC histology images. The contributions of this study are summarized as follows:1The use of the connected component analysis method to group components with similar characteristics into binary masks that assist in separating overlapping and non-overlapping objects, thus avoiding over-segmentation.2The binary masks from the connected component analysis method further aid in addressing the inaccurate segmentation of the image boundaries of intersecting objects, which is common with the active contours method. The proposed method clearly distinguishes the different ROIs from each other, clearly isolating and segmenting the cancerous lesions as visually documented in Section 3 and Section 4.

Models Inculcating active contours are preferred as a segmentation tool largely because they make it easy to separate objects of interest from the background using contours/curves. While these models can correctly capture the shapes of an object, thereby extracting requisite morphological features, most of these models fail to segment multiple structures simultaneously on large images. Therefore, most boundary-based active contour models require spot-on model initialization to handle large images [10]. While region-based models do not require accurate initialization, continuous segmentation of multiple structures on large images is still difficult, especially given a complicated image background. With this known deficiency, CAD hybrid systems have been utilized to offer solutions and or improve conventional approaches that analyze nuclei cells in BC histology images.

Research authors in [11] present a performance analysis of segmentation algorithms for computer-aided diagnosis and the detection of BC via image enhancement, nuclei segmentation, extraction, and classification. A suitable method is selected from the results of the comparative analysis and used in the proposed technique. An automated detection and segmentation method of BC nuclei objects is presented by [12], which utilizes a convolutional neural network (CNN) combined with an active contour model with adaptive ellipse fitting simultaneously. The CNN accurately detects nuclei objects and the region-based active contour model further segments nuclei objects obtained by the CNN previously initialized, and an adaptive ellipse fitting is utilized to deal with the remaining overlapping nuclei objects on the image.

Research in [13] proposes a geometric active contour model to initially detect nuclei patches in images using a trained multi-layer neural network. The pre-segmented image is then used to extract overlapping nuclei regions via the watershed algorithm, including their edges. Lastly, benign nuclei objects are identified and removed from the segmented image based on morphological characteristics. The resultant image only has cancer nuclei.

Authors in [14] qualitatively assess the performance of several active contour methods, namely, an edge-based active contour model, a geometric active contour model, and an active contour without edges method, and compare their results, settling on the one with the best results. The edge-based active contour model uses the inflation/deflation force technique that moves contours to locate edges/boundaries in an image [15]. The geometric active contour model uses curvature morphological operators to evolve contours that eventually detect boundaries/edges on images [16]. The active contour without edges [17] proposes a curve evolution-based technique that detects object boundaries, determined by the segmentation of an image rather than the desired boundary, as is the case with conventional active contour models.

Effective early diagnosis of BC requires precise identification of cancerous lesion boundaries/edges. This remains a difficult task, particularly for nonuniform regions in images. Research authors in [18] propose an active contour method that utilizes a local and global fitted function to extract unknown boundaries for regions of interest from nonuniform images. This method disregards false contours, and a bias field is introduced for both the global and local fitted models to ensure the separation of contours within an image.

New technologies also offer solutions through guidance that allows medical practitioners to view micro-structures of imaging data. In [19], augmented reality (AR) aids surgeons in seeing minute tissue structures on live video via pre-operative image data. The authors propose a mass detection system for the early detection of BC using Gaussian filters and morphology operations to pre-process image data and remove noise. The system then utilizes active contours to segment objects of interest (tumors), thus obtaining significant results that are visually clear.

Research authors in [20] propose another active contour-based approach that utilizes an extreme learning machine (ELM) method that combines image intensities and time-domain features to enhance and distinguish lesions from other tissues on an image. A robust function of the ELM combines time domain features to identify ROIs and solve the nonuniform intensities in the original image. A fuzzy function combines the Hessian shape index, time domain features, and intensities to detect irregular and blurry boundaries/edges. Hence, these two functions form an energy function that can precisely move the contours toward the boundary of the lesions; thus, BC is segmented accordingly.

A two-stage nuclei segmentation strategy [21] utilizes stain separation to isolate nuclei objects from other tissues in an image. Stain separation breeds over-segmentation, and this is a result of the overlapping objects. A marker-controlled watershed method is introduced to deal with this problem by isolating and segmenting nuclei objects with their edges. Authors in [22] group segmentation methods into classical segmentation (region, edge, threshold), machine learning segmentation (unsupervised, supervised), and deep-learning segmentation. Their findings reveal that region-based segmentation is preferred for classical methods, specifically region-growing techniques.

A novel segmentation approach [23] pre-processes images via principal component analysis and Sobel filters to gather vertical and horizontal gradient image information. A saliency map is then created from gathered gradient information, via tensor voting that highlights important nuclei features such as boundaries/edges and centers from the image background. A loopy back propagation (LBP) algorithm based on a Markov random field (MRF) is used to detect and extract nuclei boundaries, followed by the nuclei objects after further post-processing.

Research authors in [24] present a method that integrates a deep-learning framework with an improved hybrid active contour model for nuclei segmentation. Images in this method were pre-processed via color normalization, and the results were used to generate classes of training samples, namely, nuclei, background, and edges. A deconvolution operator was used to obtain non-nuclei objects, which formed the image background. After deconvolution, a CNN model was used to train the patch samples (nuclei, non-nuclei), and, to sort out over-segmentation, a third edge patch (nuclei boundaries) was used. A probability map is obtained from the CNN model and is used to form ellipses near the nuclear boundaries on the original image, thus allowing segmentation via active contours. Finally, all gathered image attributes and their intensities regarding nuclei regions, and boundary/edge information are integrated into the active contour model to assist in isolating overlapping nuclei objects from the image background.

Authors in [25] develop an automated deep-learning-based approach that utilizes the star-convex polygon method for nuclei segmentation in histology images. This method pre-processes images via color normalization and data augmentation techniques to reduce stain variability and increase dataset size, respectively. A pre-trained U-Net model is then used to train these pre-processed images, and a weight map is calculated from the isolation boundary between overlapping nuclei cell objects via various morphological operations [26]. These maps separate the background and ROI due to the semantic segmentation of the U-Net model. Therefore, individual nuclei segmentation is further handled via the star-convex polygon approach combined with the non-maximum suppression technique [27] that precisely segments individual nuclei objects with their shapes intact.

Image segmentation on low-contrast images is challenging as it requires distinct object identifiers from the background without interfering with image intensities when dealing with blurring and noisy factors during pre-processing. Hence, research authors in [28] propose a Koschmieder imaging system (KIS)-awarded active contour model to resolve low-contrast image segmentation issues. The KIS uses marker detection to enhance, estimate, and describe optical images combined with their intensity distributions after they go through the optical imaging system [29]. The low-contrast image is quantified and is the degree of variation in pixel intensities within an image, representing dark and bright regions. Image variance quantitatively assesses image contrast in the proposed model. The model is evaluated on artificially manipulated low-contrast images and its effectiveness is compared to its performance on real-world low-contrast images.

Overlapping nuclei objects are challenging when dealing with nuclei segmentation in histology images. However, various methods have been proposed to deal with the challenge. Research authors in [30] propose a solution to isolate positively stained nuclei objects from a tissue image in an immunohistochemistry (IHC) sample. They pre-process images using color deconvolution to separate the different stains, namely, positive nuclei (brown color) and negative nuclei (blue color) in the image, and gray-level co-occurrence matrix (GLCM) for texture extraction. A deep-learning approach, namely, IHC-Net, is introduced to segment positive and negative nuclei objects and the non-diagnostic regions (including overlapping objects) from the image. Consequently, a proportion score is calculated as the percentage of positive nuclei to the total number of positive and negative nuclei.

BC tumors have a high risk of metastasizing, thus spreading to other parts of the body. Hence, early detection of such metastatic regions can be handled by predicting the growth rate of these BC tumors in histology images. Authors in [31] propose a novel Morpho-Contour Exponential Estimation (MoCEE) method that utilizes an enhanced mask region-based convolution neural network (R-CNN) combined with active contours to ensure accurate BC tumor segmentation on magnetic resonance imaging (MRI) datasets.

Firstly, images are pre-processed via morphology operations to reduce noise and highlight both salient tumor features and regions on the image. These salient features include texture, intensity variations, and tumor boundaries/edges, while the ROIs include regions with unclear boundaries, indicating the presence of cancer or metastasis. These morphological features are then integrated into the masked RNN via an active contour function based on the image properties to allow further enhanced segmentation.

The active contour function primarily segments the breast tumor and analyzes the distance between two random points on the segmented region to determine the growth rate. The gradient-boosting and exponential models are then used to predict tumor growth from the derived features via iterative learning of feature vectors for better prognosis. The gradient-boosting approach iteratively predicts tumor size to approximate the true growth rate.

Most of the discussed methods that inculcate the active contours method have failed to deal with the issue of edge/boundary inhomogeneity, specifically when dealing with intersecting/overlapping objects. Therefore, this study proposes a hybrid active contour segmentation method that combines with the connected components analysis method to isolate, accentuate, and segment cancerous lesions in unsupervised breast histological images.

## 2. Proposed Segmentation Method

This proposed method for segmenting BC histology images utilizes a combination of the capabilities of both active contours and connected component analysis. Figure 1 shows the proposed hybrid method and is summarized by the steps below.

### 2.1. Dataset Pre-Processing

The dataset used in this study was obtained from the Kaggle repository and has 24 unsupervised hematoxylin and eosin (H&E) images. The dataset is of limited size and, therefore, unsuitable for processing by deep-learning neural networks. To solve this problem, data augmentation is utilized to increase the dataset size and allow deep learning.

#### 2.1.1. Dataset Augmentation

There has been an issue with a limited number of medical images in publicly available datasets, making it impossible for neural networks to make intelligent decisions [32]. A key pre-processing method that is effective in training highly discriminant deep-learning models is data augmentation [33].

The authors in [34], propose data augmentation as their preferred method for increasing dataset size in specific fields with limited publicly available datasets, and it is often used in medical image processing. Accessibility of large medical image datasets is difficult due to ethical concerns and labeling costs. Hence, data augmentation greatly increases the size and variety of data available for training, without collecting new samples [35].

Deep-learning approaches require huge amounts of data to train and test models to achieve good evaluation performance. Data augmentation is therefore utilized, which refers to artificially creating images through different transformations, thus increasing the dataset size. These augmentation methods range from simple, effective transformations such as cropping, flipping, padding, width and height shifts, rotation, flipping, and scaling.

In this proposed study, 24 unsupervised BC histology images are augmented through various transformations, namely, height and width image shifts, scaling, and rotation arguments to produce 15,179 images. Data augmentation through patch rotation and mirroring improves and increases dataset size without necessarily worsening the quality [36]. Consequently, the problem herein is rotation invariant, which means that various pathologists can analyze BC histology images in different orientations without altering the diagnosis.

#### 2.1.2. Data Stain Normalization

During the preparation and pre-processing stages, images undergo various distortions and inconsistencies. These are attributed to the H&E staining procedure used to magnify minute nuclear features, tissue structures, and image transformations resulting from data augmentation. Laboratory slide preparation, examination, analysis, and digitalization of scanning samples are other factors that lead to image variations [37]. These factors negatively impact the training and testing of neural networks. Hence, a stain normalization technique is needed to remove color irregularities in medical images and improve the model’s efficacy. In this proposed study, the Macenko et al. [38] stain normalization technique is used for BC images to separate nuclei features from tissue structures. Images in the dataset are first converted from the BGR to the RGB color space to enable smooth stain normalization.

Macenko stain normalization: is used to prepare tissue slides. Image colors are converted to their optical density (OD) equivalent via a simple logarithmic transformation,$$OD=-\log_{10}\left(I\right)$$

With *I* as the RGB color vector and individual components normalized to [0, 1].

A value β, is used as a threshold value to remove data with higher OD intensity.

Single value decomposition (SVD) is applied to optical density tuples to create a plane. The plane corresponds to the largest singular values. OD-transformed pixels are then projected onto the plane to determine the angle at each point related to the first SVD direction. The color space transformation is applied to the original BC histology image. An image histogram is stretched such that the range covers the lower (100−β)% of the data.

Minimum and maximum vectors are calculated and projected back to the OD space. The hematoxylin stain corresponds to the minimum vector while the eosin stain is the maximum vector. Stain concentrations are determined to form a matrix representing RGB channels and OD intensities, respectively. This study has values of α and β set at 1 and 0.15, respectively. Figure 2 shows image results from the entire data pre-processing of BC histology images, respectively. After this stain normalization process, our proposed approach focuses on the normalized H image (image with only nuclei objects), which has been isolated from the greater normalized H&E image set.

### 2.2. Image Enhancement

Enhancing dataset images improves their brightness, contrast, and scaling to compensate for the non-uniformity resulting from image illumination. This proposed study handles this step through thresholding, morphology operations, and distance transform.

#### 2.2.1. Thresholding

Binary thresholding is used to capture the outline of the BC ROI on the normalized histology image/images and is shown by Figure 3.

#### 2.2.2. Morphology Operations

These include dilation and erosion operations to remove noise, remove overlapping edges, and extract certain regions on the BC histology images. Opening and closing operations are used to distinguish between the background and the foreground in an image. These regions are differentiated by diminishing and accentuating image pixels and edges. These operations also highlight the unknown area between the background and the foreground. Figure 4 shows images resulting from morphology operations.

#### 2.2.3. Distance Transform

This method isolates nuclei objects on the image and finds the sure foreground by removing the remaining uninteresting ROIs. It also highlights and emphasizes the sure foreground objects and the sure background on the BC histology image. Figure 5 shows the image after distance transformation.

### 2.3. Segmentation

Nuclei segmentation is handled through a combination of both the connected components analysis method and the active contours segmentation method.

#### 2.3.1. Connected Component Analysis

The CCA method combines pixel components with neighborhoods with the same properties. Additionally, image pixels are grouped into connected darker and brighter regions. Darker regions form the background, while brighter pixels form the foreground. The connected component analysis (CCA) method extracts these ROIs as binary masks on BC histology images. These binary masked ROIs are shown in Figure 6.

The detection of darker and brighter regions is influenced by (*LoG*) Laplacian of Gaussian approach, a convolution kernel of the form:(1)$$LoG=\frac{x^{2}+y^{2}-2\sigma^{2}}{\sigma^{4}}e^{-\frac{x^{2}=y^{2}}{2\sigma^{2}}}$$
such that σ is the kernel width.

#### 2.3.2. Active Contours Segmentation

Active contours have been widely used in the segmentation of medical images, targeting computer vision tasks that describe the boundaries of shapes in an image. They are particularly utilized to solve cases where the approximate shape of a boundary/edge is known. The active contours deformed (snake) model characteristic adapts and evolves to image color variations, matching, and tracking of object boundaries/edges. The approach also finds elusive boundaries (contours) shapes by ignoring missing boundary/edge information.

Research authors in [39] propose an active contour-based model (snake) to detect boundaries of objects from deformed initial contours. The deformed contours use an energy function that decreases when the snake perfectly fits the object boundary in an image. With the increased number of objects in various medical images, the snake approach finds it difficult to segment images. Therefore, numerical step-by-step procedures proposed by [40,41] are used to detect topology changes automatically on the image.

The resultant binary-masked images from the previous connected components analysis method are then used for contour detection. Multiple boundaries of objects are detected as shown by the equation below:

Let $C\left(p,t\right):0,1\to R^{2}$ denote a family of curves resulting from the motion $C_{0}\left(P\right)$ directed towards inward Euclidean vector *N*. Let *I* denote the image where object boundaries are to be identified and detected. We assume the plane evolution of the curve is given by(2)$$dCdt=g\left(I\right)\left[k+v\right]N,C\left(p,t=0\right)=C_{0}\left(initialcurve\right),$$
where *v* is a constant, *k* local curvature, *N* unit vector normal to curve, and g(I) a factor related to image content.

We assume the deforming curve C(p,t) is the zero value of a function ⋃, that is C(p,t) a set of points (x,y,t) given U(x,y,t)=0. Given Equation (Equation 2) and the derivative of U(x,y,t)=0 with respect to space and time, the deformation of C(p,t) is given by deformation of surface ⋃(x,y,t) whose evolution is given by(3)$$dU\left(x,y,t\right)dt=g\left(I\right)dvi\nabla ⋃|⋃|+v|\nabla ⋃|,⋃\left(x,y,t=0\right)={⋃}_{0}\left(x,y\right),$$

Given that |∇⋃| denotes the magnitude of the gradient, dvi denotes the divergence operator, and ${⋃}_{0}$ is the level set representation of $C_{0}$.

Consequently, authors in [42,43] presented the aspect of geodesic active contours that results in a geometric model given by: (4)$$dU\left(x,y,t\right)dt=g\left(I\right)dvi\nabla ⋃|⋃|+v|\nabla ⋃|,+\nabla g\nabla ⋃⋃\left(x,y,t=0\right)={⋃}_{0}\left(x,y\right).$$
and from these results, a field is generated ⋃(x,y,t) having null positions corresponding to active contour locations at any given evolution time. In these equations, *v* is a constant that constrains active contours from either expanding or shrinking and is a function of the method used to draw the initial coarse contour. This constant is key within the model because it allows the initial curve to gather a non-convex shape. dvi(∇⋃/|⋃|) denotes the curvature of the level set passing by a point and determines the regularising effect of the model.

The function g(I) is utilized as a stopping factor of the evolving curve, namely, the factor is small near an edge/boundary to stop evolution when the contour gets close to the edge. The function g(I) is expressed as:$$g\left(I\right)=1(1+|\nabla \left(I_{1}{|}^{p}\right),$$

Given $I_{1}$ results from the low-pass Gaussian filtering of image I,p=1or2, and other expressions of g(I) can be used to monitor other features. The effect of ∇(g)∇⋃ is towards capturing the evolving contour as it moves towards an edge and pushes it back if it crosses the edge. Therefore, unlike the conventional snake method, the geometric active contour model is stable and handles topographical changes, namely, the splitting and merging, devoid of any computational problems.

The effect of the proposed hybrid active contours segmentation method on unsupervised BC histology images was evaluated using a deep-learning recurrent neural network model. The recurrent neural network architecture consisted of eight layers: one input dense layer, three hidden dense layers, one output dense layer, and three dropout layers.

Several rounds of evaluations via the recurrent neural network were undertaken while tweaking the cross-entropy, learning rate, epochs, and dropout to produce significant results that reduce over-fitting. The optimal parameters that achieved these in our iterative experiments included minimizing cross-entropy using categorical cross-entropy with the Adam optimizer, a learning rate of 0.0001, a batch size of 32, a dropout of 0.2, and 30 epochs. We settled on the hyperparameters above from iterative model experiments.

## 3. Results and Discussion

The experimental results of this study are based on the performance analysis of the ACCA segmentation method on unsupervised BC histology images. Experiments were carried out on 15,179 unsupervised BC histology images split into 11,384 training and 3795 testing set images. The processing of histology images from the unsupervised BC dataset to aid data analysis has been discussed in the previously proposed method section. The proposed method is also executed by Algorithm 1.
**Algorithm 1** Proposed segmentation method for unsupervised breast cancer histology images segmentation  1:**procedure **Encoder(Y)         ▹$Y:y_{i}$ is an input image with dimension (L, H).  2:    Extract the feature map $Feature_{map}$ from the input image  3:    Initialization: ${\mathrm{Feature}}_{i}=y_{0}$;  4:    **for** (i=0:N−1) **do**  5:        ${\mathrm{AugmentedFeature}}_{i}=\mathrm{DataAugmentation}\left({\mathrm{Feature}}_{i}\right)$;  6:        ${\mathrm{GrayScalingFeature}}_{i}=\mathrm{GrayScaling}\left({\mathrm{AugmentedFeature}}_{i}\right)$;  7:        ${\mathrm{FilteringFeature}}_{i}$;  8:    **end for**  9:**end procedure**10:**procedure **Procedure Enhancement(${\mathrm{GrayScalingFeature}}_{i}$)       ▹ It is the output of the preprocessing procedure11:    **for** (j=0:M−1) **do**12:        ${\mathrm{ErisonFeature}}_{i}=\mathrm{ErisonMorophology}\left({\mathrm{FilteringFeature}}_{i}\right)$;13:        ${\mathrm{DilationFeature}}_{i}=\mathrm{DilationMoropology}\left({\mathrm{GrayScalingFeature}}_{i}\right)$;14:    **end for**15:**end procedure**16:**procedure** Procedure Segmentation(${\mathrm{DilationFeature}}_{i}$)       ▹ It is the output of the enhancement procedure17:    **for** (s=0:L−1) **do**18:        ${\mathrm{ThresholdingFeature}}_{i}=\mathrm{Thresholding}\left({\mathrm{DilationFeature}}_{i}\right)$;19:        ${\mathrm{ConnectedFeature}}_{i}=\mathrm{ConnectedComponentAnalysis}\left({\mathrm{ThresholdingFeature}}_{i}\right)$;20:        ${\mathrm{ActiveContoursFeature}}_{i}=\mathrm{ActiveContours}\left({\mathrm{ConnectedComponentFeature}}_{i}\right)$;21:    **end for**22:**end procedure**23:**procedure** Procedure FeatureExtraction(${\mathrm{ActiveContoursFeature}}_{i}$)       ▹ It is the output of the segmentation procedure24:    **for** (r=0:K−1) **do**25:        ${\mathrm{Feature}}_{i}=\mathrm{FeatureExtraction}\;\left({\mathrm{ActiveContoursFeature}}_{i}\right)$;26:        Segmented Image Feature= ${\mathrm{Feature}}_{i}$;       ▹ Final Segmented Output Display.27:    **end for**28:**end procedure**

Figure 1 shows an overview of all processing stages within the proposed method, such as pre-processing, image enhancement, segmentation, and feature extraction. Additionally, the performance evaluation of the proposed approach is based on its automatic ability to segment cancerous lesions on breast histology images, as shown by Figure 7.

The practicability of our approach has been tested on the publicly available Warwick QU dataset from the Kaggle dataset repository. Figure 8 shows original images from the dataset above that consist of other human gland histology images, namely, the liver, colon, and kidney, just to name a few, and are available publicly. Figure 9 shows the output images after our method was utilized on the augmented images from the original images due to the small dataset size. Figure 10 shows the final output masked images with contours marked after our method was utilized. Figure 11 illustrates the training and validation graph curves of the Warwick QU dataset after applying the proposed segmentation results, which yielded a significant 95.73% accuracy score.

Table 1 shows the accuracy results from the performance evaluation carried out on the augmented BC histology images at the preprocessing stage. The result is then compared to the accuracy results after the proposed method is evaluated on the final segmented images, which shows an improvement in evaluation performance.

Our proposed technique achieved significant results in identifying and highlighting BC lesions throughout the unsupervised histology image dataset. These results are attributed to various reasons, namely dataset augmentation that increases the dataset size, and stain normalization, which reduces image color inconsistencies. Table 2 shows the comparison between our proposed approach and other state-of-the-art methods related to BC human histology images. The table highlights methods used in specific architectures targeting nuclei segmentation, which have been previously discussed in the related work section. The efficiency of how these methods have been able to segment nuclei objects on histology images is thus compared to the proposed method in this study via performance evaluation accuracy results.

Figure 12 illustrates the proposed method’s training and validation loss/accuracy graph curves. The graph curve shows the best fit curve after several simulations of the proposed model. The optimal results, visually illustrated, are after hyperparameter tuning, namely, epochs 30, learning rate 0.0001, categorical cross-entropy, and the Adam optimizer. The graph shows a reduction in over-fitting due to early stopping at 30 epochs, and a dropout rate of 0.2.

### Limitations

The availability of public datasets, specifically those targeting medical imaging, is hard to come by, mostly because of ethical concerns.

Most active contour methods also used in medical imaging segmentation fail to identify, isolate, and detect multiple tissue structures found on large images.

Active contours also fail to resolve image boundaries of intersecting and overlapping nuclei objects, thus hindering the absolute identification and isolation of nuclei objects on histology images. Additionally, most existing nuclei segmentation methods tend to segment nuclei objects on the entire normalized image, which contains both nuclei features and tissue structures, instead of just focusing on the nuclei features ROI, thus increasing processing time.

Another limitation is the reluctance of pathologists to invest in CAD systems due to their high number of false positive results. Hence, there is a need to introduce deep-learning neural networks within CAD systems to assist in the early diagnosis and treatment of BC. Therefore, the proposed method of this study uses the efficiency of the connected component analysis method to specifically highlight and isolate ROIs, thus solving limitations of the active contour method. This proposed method also segments cancerous lesions on nuclei objects within human breast histology images. This segmentation method targets isolating the nuclei features ROI from other tissue structures, from the larger normalized image, forming a normalized H (nuclei features) dataset image. Hence, this leads to easier and faster model processing.

## 4. Conclusions

Most of the methods discussed in the introduction section and related to the method proposed tend to focus on one of the methods, either active contours or connected components. These methods individually have their bottlenecks. Active contours do not resolve edge inhomogeneity, while connected components analysis does not resolve intersecting objects, thus over-segmenting.

Hence, this proposed study discusses implementing the ACCCA segmentation method of cancerous lesions on unsupervised BC histology images. It breaks it down into pre-processing, data augmentation, image enhancement, and segmentation stages. Data augmentation is used to increase the limited dataset size. Stain normalization removes color inconsistencies and irregularities resulting from the data augmentation process. Thresholding and morphology operations enhance and highlight BC lesions with their edges on histology images; the connected components method groups components with similar characteristics into binary masks.

The binary masks are then used within the RNN to separate overlapping and non-overlapping objects. Lastly, the active contours method uses the resultant binary masks from the CCA method to resolve image boundaries/edges by segmenting, maintaining, and topographically distinguishing nuclei objects and their edges on different image regions, namely, background and foreground.

An accuracy performance evaluation protocol is used to evaluate the efficiency of the proposed model in meeting its segmentation task. The proposed method produces significant results as compared to other state-of-the-art methods. Specifically, it utilizes the connected component analysis method to group objects with similar properties, thus resolving edge inhomogeneity, which is common with the active contours method. Therefore, the proposed method separates intersecting objects and highlights edges/boundaries, segmenting cancerous lesions in unsupervised BC histology images.

From the literature reviewed herein, our study has made several observations that are key in the segmentation and detection of breast cancer lesions in histology images. Pre-processing images through data augmentation is important in cases of scarce publicly available datasets by increasing the dataset size. Stain normalization is necessary for removing color irregularities on images caused by data augmentation and assists in nucleus identification, depending on the initial staining procedure. Thresholding and morphological operations are key in improving the quality of the image by highlighting and accentuating nuclei objects, and opening and closing operations are mostly preferred.

High false-positives and false-negatives brought about by the active contours method are usually handled best by the choice made for image pre-processing. Notably, the image pre-processing and morphology operations used in a study will decide the segmentation method to use, thus offering a ripple effect on the final performance of the model.

## 5. Future Work

Therefore, there is a need for automatic early diagnostics systems in place of manual methods to assist in easier and faster diagnosis and to act as a second opinion for clinicians to detect BC lesions in histology images. This proposed study provides great solutions for learning models to correctly identify, isolate, segment, and detect nuclei objects and their edges on BC histology images. Despite the significant outcome of this proposed segmentation method with supervised models, a series of encouraging future perspectives sprout, namely, the availability of public datasets, the introduction of blended approaches, and the inclusion of weight regularization methods fine-tuned for unsupervised models, thus increasing model efficacy and reducing over-fitting. These avenues are discussed in detail as follows:**Data availability and integrity**—most deep-learning approaches require significantly huge datasets to deduce meaningful and effective performance results. Therefore, it is necessary to access more publicly available BC histology image datasets, thus aiding deep learning. Additionally, the proposed model should be tested on other huge volume datasets to evaluate performance, not those specifically targeting breast cancer.**Regularization methods**—to improve the performance of models. This can be done through model hyperparameter tuning, such as optimizing learning rates, dropout, loss functions, activation functions, and early stopping methods.**Blended approaches**—combining various/several methods and their attributes to form hybrid methods, thus improving overall evaluation performance. This amalgamation can occur at any stage of the model architecture, namely, pre-processing, combining various attributes of different models to form one that will enhance the training, extraction, detection, and classification of nuclei objects. Additionally, in the future, our work can expand to infiltrate and diagnose image datasets of other human/animal gland histology images, not just limited to BC histology images.Consequently, the segmentation accuracy of these medical conditions could also be boosted by including attention-based models and other deep and machine-learning techniques. Results from the performance evaluation of these models could then be used to build clinical trust and thus the utilization of the proposed models. Furthermore, we plan to explore other models, including but not limited to CNN, LSTM, and UNet.

## Figures and Tables

**Figure 1 bioengineering-12-00642-f001:**
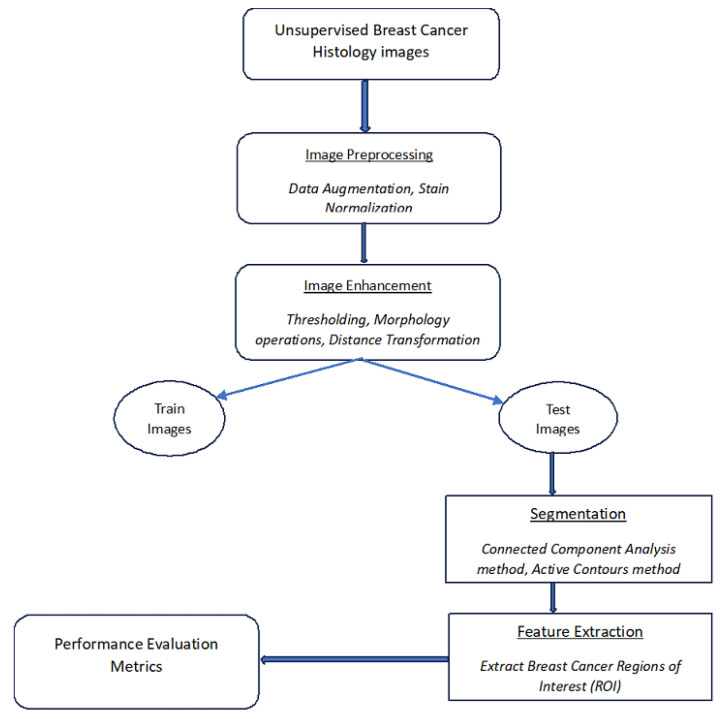
Flow diagram of proposed segmentation method.

**Figure 2 bioengineering-12-00642-f002:**
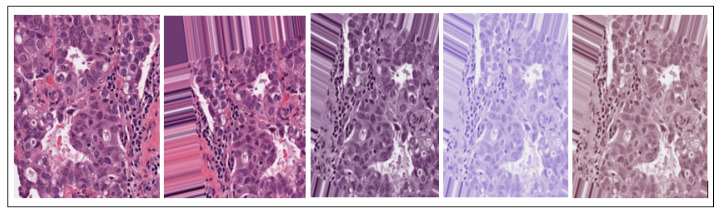
Pre-processed images, namely, Original, Augmented, Normalized H&E, Normalized H, and Normalized E breast cancer histology images, respectively.

**Figure 3 bioengineering-12-00642-f003:**
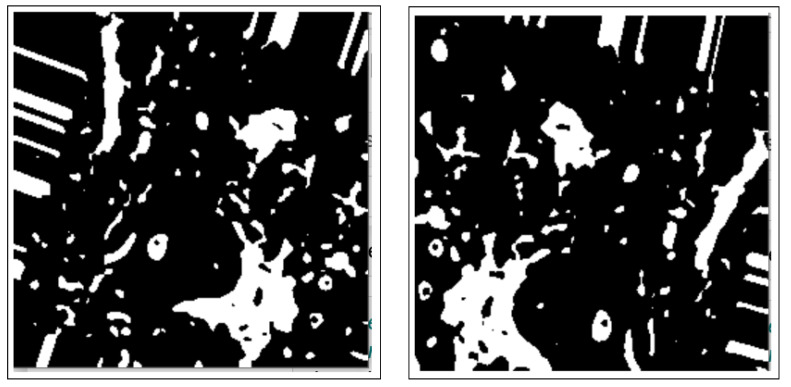
Images after binary thresholding.

**Figure 4 bioengineering-12-00642-f004:**
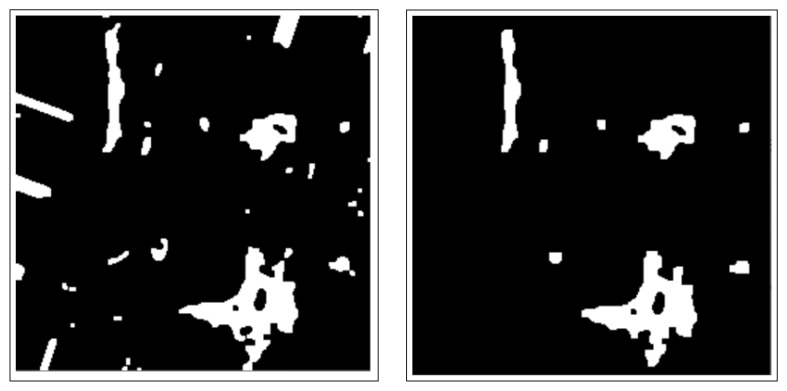
Images after morphology operations namely; erosion and dilation operations, opening operation (clearing borders), respectively.

**Figure 5 bioengineering-12-00642-f005:**
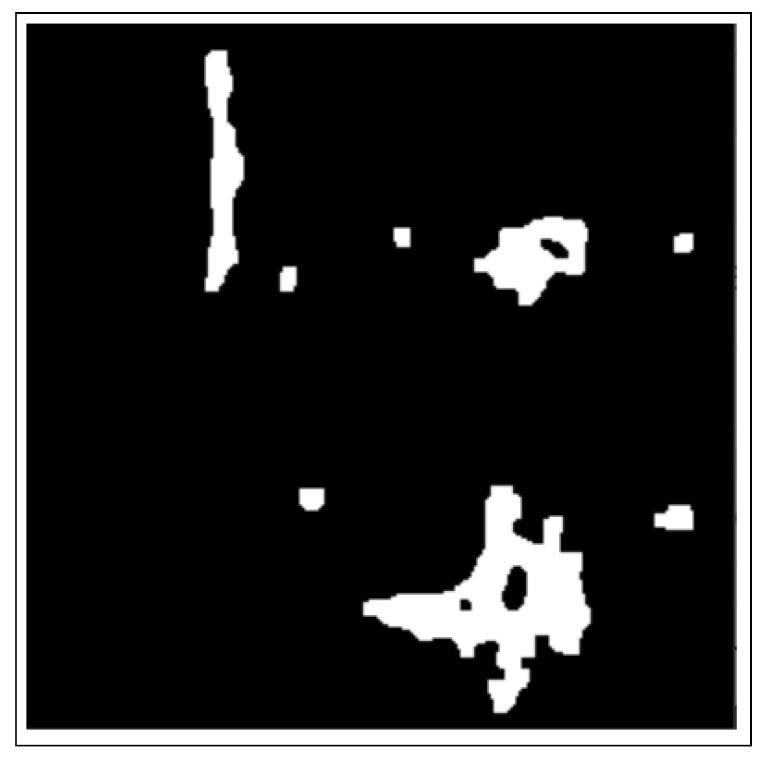
Image after distance transformation.

**Figure 6 bioengineering-12-00642-f006:**
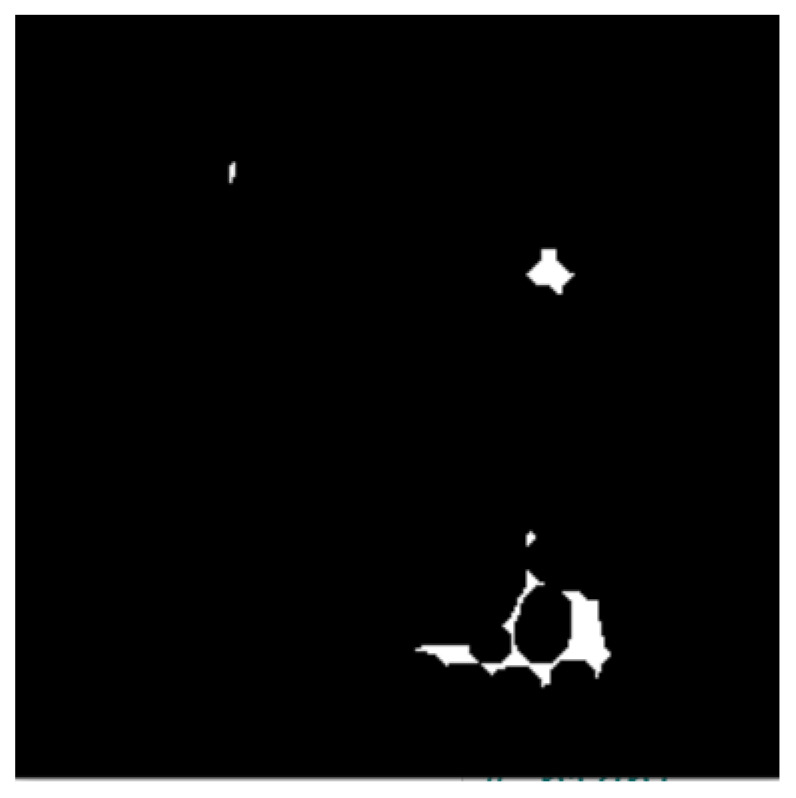
Image after connected component analysis.

**Figure 7 bioengineering-12-00642-f007:**
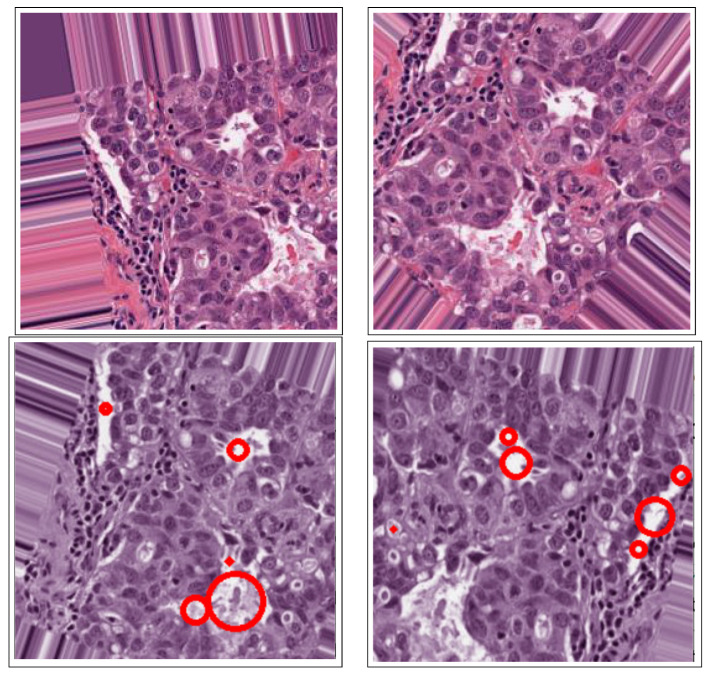
(**First row**): Original dataset images, (**second row**): Corresponding image results from the proposed segmentation method.

**Figure 8 bioengineering-12-00642-f008:**
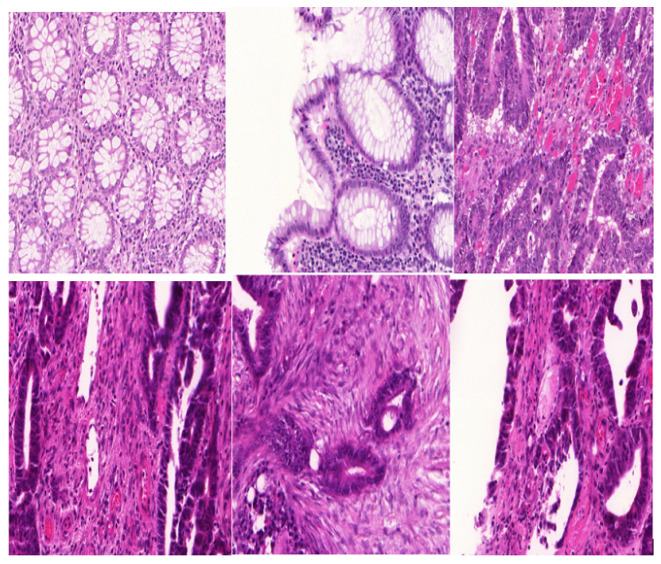
Original images of the Warwick QU dataset.

**Figure 9 bioengineering-12-00642-f009:**
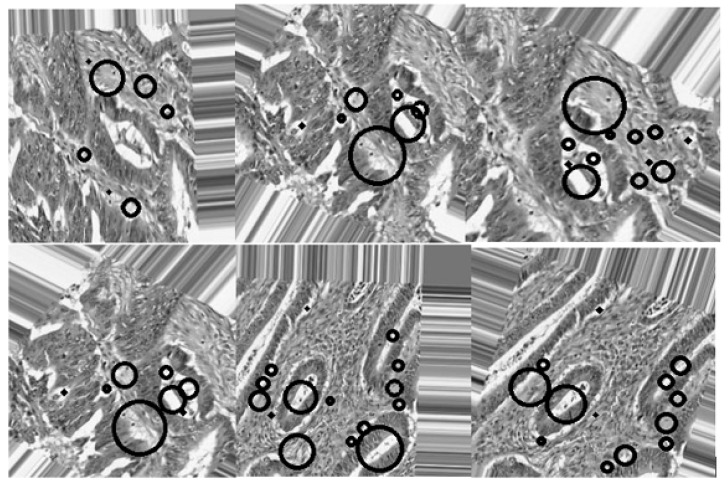
Image results from the proposed segmentation method on original augmented images from the Warwick QU dataset.

**Figure 10 bioengineering-12-00642-f010:**
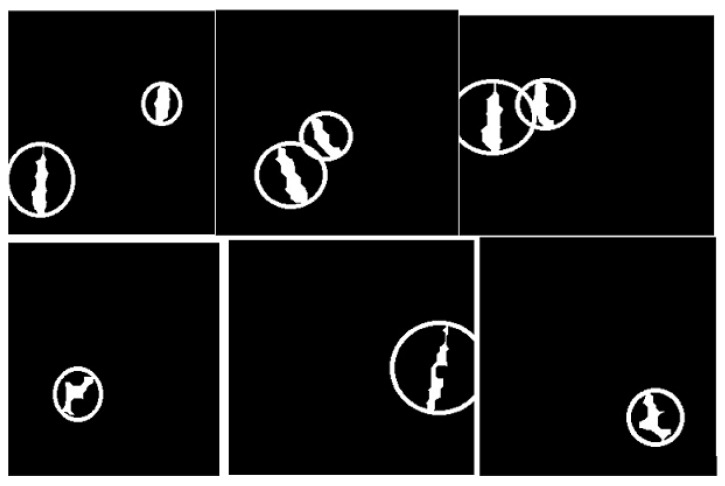
Final masked image results after proposed segmentation method was applied on images from the Warwick QU dataset.

**Figure 11 bioengineering-12-00642-f011:**
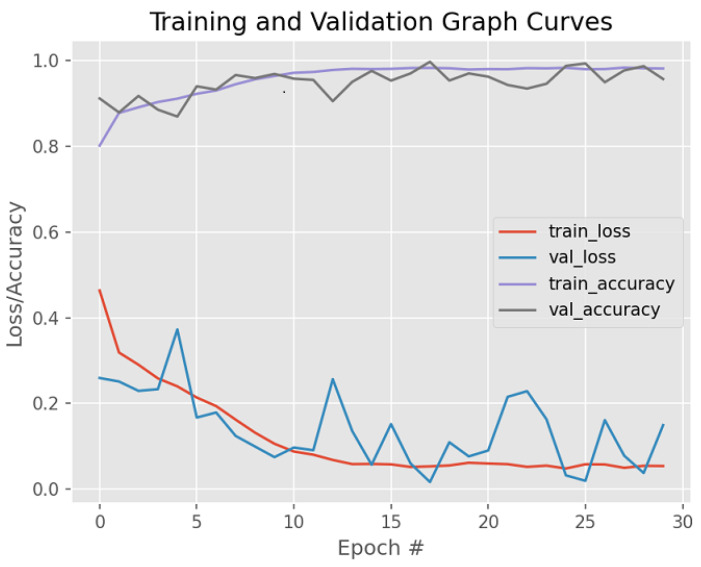
Graphical image results on the Warwick QU dataset after active contour connected component analysis.

**Figure 12 bioengineering-12-00642-f012:**
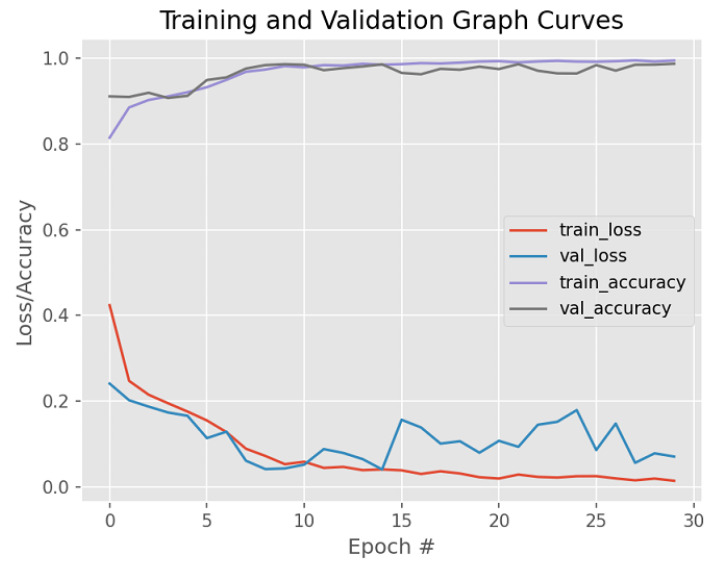
Graphical image after active contour + connected component analysis.

**Table 1 bioengineering-12-00642-t001:** Performance evaluation results before and after applying ACCCA segmentation method.

Dataset	Augmented Images	After Implementation of ACCCA Method
**Accuracy**	**94.0%**	**98.71%**

**Table 2 bioengineering-12-00642-t002:** Comparison table of the proposed method to other state-of-the-art methods related to BC histology images.

Reference Authors	Methods	Accuracy
Mahanta et al. [30]	Deep-learning-based nuclei segmentation + ensemble classification scheme	98.24%
Ronneberger et al. [26]	U-Net	92.03%
Fatakdawala et al. [10]	Expectation max driven geodesic active contours with overlap resolution	86%
Xu et al. [12]	CNN active contour model with adaptive ellipse fitting	85.71%
Mouelhi et al. [13]	Colour active contour model + improved watershed method	97%
Niaz et al. [18]	Inhomogeneous image segmentation using hybrid active contour model	98.3%
Kaladevi et al. [31]	Morpho-contour exponential estimation algorithm	93.12%
Hu et al. [21]	Two-stage nuclei segmentation strategy	92.5%
Paramanandam et al. [23]	Tensor voting + Loopy Back Propagation algorithm	93.0%
Zhao et al. [24]	Deep CNN + active contour method	85.0%
Nelson et al. [25]	Star-convex polygon approach + non-maximum suppression technique	66%
**Proposed Method**	**Active contours + connected components analyis (ACCCA)**	**98.71%**

## Data Availability

The data and code used to support the findings of this study can be obtained from the corresponding authors upon request. The data are not publicly available due to [ethical concerns and limited publicly available datasets].

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
