# Peer review of "Active Contours Connected Component Analysis Segmentation Method of Cancerous Lesions in Unsupervised Breast Histology Images"

_bioengineering, 2025, doi:10.3390/bioengineering12060642_

Round 1
Reviewer 1 Report (Previous Reviewer 4)
Comments and Suggestions for Authors
Dear authors,
the manuscript is interesing however, some minor modifications are required to improve the manuscripts quality.
- In the intro section please explicitly state the hypotheses of your research in the bullet format
- Related works should be inside the intro section. And based on other research papers you should define the disadvantages and what is the novelty of your research.
- In line 417 of the manuscript you mention F1 accuracy, The accuracy is one evaluation metric and the F1_score is the other evaluation metric. Please clear that.
Author Response
Manuscript ID: bioengineering-3450283
Title: Active Contours Connected Component Analysis segmentation method of cancerous lesions in unsupervised breast histology images
To: Bioengineering
Re: Response to reviewers
REVIEWER 1
Concern 1
In the intro section please explicitly state the hypotheses of your research in the bullet format.
Authors’ Response:
We thank the reviewer for pointing this out. This point has been addressed in the revised manuscript. The specific has been addressed in the introduction section and, is highlighted in Blue in the paper.
The main contributions of this paper are to resolve intersecting nuclei objects and their edges’ inhomogeneity. Hence, assisting in correctly identifying, isolating, segmenting, and detecting cancerous lesions on BC histology images.
Action required: The above paragraph has been added to the introduction section and, is highlighted in Blue in the paper.
Concern 2
Related works should be inside the intro section. And based on other research papers you should define the disadvantages and what is the novelty of your research.
Authors’ Response:
We thank the reviewer for pointing this out. The specific concern has been addressed at the end of the introduction section. This is highlighted in Blue in the paper.
Most of the discussed methods that inculcate the active contours method have failed to deal with the issue of edge/boundary inhomogeneity, specifically when dealing with intersecting/overlapping objects. Therefore, this study proposes a hybrid active contours segmentation method for contour segmentation method that combines with the connected components analysis method to isolate, accentuate, and segment cancerous lesions in unsupervised breast histology images and is discussed meticulously in the leading section histological images.
Action required: The above paragraph has been added to at the end of the introduction section. This is highlighted in Blue in the paper.
Concern 3
- In line 417 of the manuscript you mention F1 accuracy, The accuracy is one evaluation metric and the F1_score is the other evaluation metric. Please clear that.
Authors’ Response:
The specific has been addressed in the results and discussion section. This is highlighted in Blue in the paper

Reviewer 2 Report (New Reviewer)
Comments and Suggestions for Authors
The title of the manuscript is appropriately informative and accurately reflects the core technique applied in the research. It clearly communicates the integration of active contours and connected component analysis (ACCCA) for the segmentation of cancerous lesions in breast histology images. The abstract is succinct and effectively summarizes the motivation behind the study, the methodology adopted, and the results achieved, including a notably high F1-score of 98.71%, which is a strong indicator of the model’s performance.
The manuscript introduces a hybrid segmentation method referred to as ACCCA, which stands for Active Contours with Connected Component Analysis. This approach targets the accurate identification of nuclei in unsupervised breast cancer histology images. The motivation is well grounded, as traditional Active Contour Models (ACMs) are known to struggle with segmenting overlapping objects. The proposed use of CCA helps to overcome this issue by better initializing and separating regions of interest. Among the major contributions of the paper are the effective integration of morphological preprocessing with the ACCA framework, the application of stain normalization to reduce color inconsistencies, and the incorporation of a recurrent neural network (RNN) to refine segmentation results from binary masks. The model’s F1-score surpasses many other state-of-the-art methods, highlighting the significance of the contributions.
The methodology presented in the paper is structured in a logical and systematic manner. The pipeline begins with preprocessing, where data augmentation techniques increase the dataset from a mere 24 images to over 15,000, followed by stain normalization using the Macenko method to reduce color variance. The next stage involves image enhancement using various morphological operations and distance transformation to emphasize boundaries. Finally, segmentation is performed, where connected component analysis generates binary masks that are further refined by active contour modeling, guided by an RNN. The algorithm is modular and is explained through pseudocode and sequential steps, making it relatively accessible. However, certain portions, particularly the equations related to contour evolution, are complex and could be better understood with the help of clearer illustrations or flowcharts.
In terms of experimental design, the authors use a small publicly available dataset from Kaggle, which they enhance through extensive augmentation. This is a valid approach given the scarcity of annotated medical image datasets. The evaluation metrics include F1-score and accuracy, both of which are clearly reported. The results demonstrate that the proposed method outperforms most existing techniques, and the authors support their claims with comparative tables and graphical visualizations that show improved outcomes in lesion segmentation.
There are several notable strengths in this study. The method combines classical image processing techniques with deep learning in a meaningful way. Its applicability to real-world scenarios is high, especially considering the unsupervised nature of the data. The paper includes a thorough comparison with existing state-of-the-art methods, providing clear evidence of improved performance. Moreover, the preprocessing pipeline is carefully designed to manage variability in H&E stained slides, which is a critical factor in histopathological image analysis.
Despite its strengths, the paper does have areas that need improvement. The overall clarity and language quality would benefit significantly from professional proofreading. There are multiple grammatical errors and instances of awkward phrasing that detract from the readability of the manuscript. Furthermore, the mathematical content in key sections is presented in a dense and technical manner, making it difficult for readers without a strong background in mathematical modeling to follow. The choice of a recurrent neural network (RNN) for image segmentation tasks is also questionable, as convolutional neural networks (CNNs) are more commonly and effectively used for spatial data. Providing a rationale or comparative analysis for this choice would strengthen the methodology. Another limitation is the narrow scope of the dataset, which lacks diversity in terms of staining protocols and image types. Although augmentation is used, it does not fully guarantee the generalizability of the model. Additionally, the absence of publicly shared code or implementation details reduces the reproducibility of the study, which is a key aspect in academic research.
Looking ahead, the paper points to several promising directions for future work, which align with the suggestions that emerge from the review. The authors could expand their testing to include other publicly available datasets, such as MoNuSeg, to validate the robustness of their approach. There is also potential to apply the method to histological images from other organs or with different staining techniques. Exploring modern machine learning architectures such as transformer-based models or attention mechanisms could further improve segmentation accuracy. Lastly, incorporating features like model explainability or confidence estimation could increase the clinical trust and usability of the segmentation results.
Author Response
Manuscript ID: bioengineering-3450283
Title: Active Contours Connected Component Analysis segmentation method of cancerous lesions in unsupervised breast histology images
To: Bioengineering
Re: Response to reviewers
REVIEWER 2
Concern 1
The title of the manuscript is appropriately informative and accurately reflects the core technique applied in the research. It clearly communicates the integration of active contours and connected component analysis (ACCCA) for the segmentation of cancerous lesions in breast histology images. The abstract is succinct and effectively summarizes the motivation behind the study, the methodology adopted, and the results achieved, including a notably high F1-score of 98.71%, which is a strong indicator of the model’s performance.
The manuscript introduces a hybrid segmentation method referred to as ACCCA, which stands for Active Contours with Connected Component Analysis. This approach targets the accurate identification of nuclei in unsupervised breast cancer histology images. The motivation is well grounded, as traditional Active Contour Models (ACMs) are known to struggle with segmenting overlapping objects. The proposed use of CCA helps to overcome this issue by better initializing and separating regions of interest. Among the major contributions of the paper are the effective integration of morphological preprocessing with the ACCA framework, the application of stain normalization to reduce color inconsistencies, and the incorporation of a recurrent neural network (RNN) to refine segmentation results from binary masks. The model’s F1-score surpasses many other state-of-the-art methods, highlighting the significance of the contributions.
The methodology presented in the paper is structured in a logical and systematic manner. The pipeline begins with preprocessing, where data augmentation techniques increase the dataset from a mere 24 images to over 15,000, followed by stain normalization using the Macenko method to reduce color variance. The next stage involves image enhancement using various morphological operations and distance transformation to emphasize boundaries. Finally, segmentation is performed, where connected component analysis generates binary masks that are further refined by active contour modeling, guided by an RNN. The algorithm is modular and is explained through pseudocode and sequential steps, making it relatively accessible. However, certain portions, particularly the equations related to contour evolution, are complex and could be better understood with the help of clearer illustrations or flowcharts.
In terms of experimental design, the authors use a small publicly available dataset from Kaggle, which they enhance through extensive augmentation. This is a valid approach given the scarcity of annotated medical image datasets. The evaluation metrics include F1-score and accuracy, both of which are clearly reported. The results demonstrate that the proposed method outperforms most existing techniques, and the authors support their claims with comparative tables and graphical visualizations that show improved outcomes in lesion segmentation.
There are several notable strengths in this study. The method combines classical image processing techniques with deep learning in a meaningful way. Its applicability to real-world scenarios is high, especially considering the unsupervised nature of the data. The paper includes a thorough comparison with existing state-of-the-art methods, providing clear evidence of improved performance. Moreover, the preprocessing pipeline is carefully designed to manage variability in H&E stained slides, which is a critical factor in histopathological image analysis.
Authors’ Response:
Thank you for the encouraging remarks.
Action required: None.
Concern 2
Despite its strengths, the paper does have areas that need improvement. The overall clarity and language quality would benefit significantly from professional proofreading. There are multiple grammatical errors and instances of awkward phrasing that detract from the readability of the manuscript. Furthermore, the mathematical content in key sections is presented in a dense and technical manner, making it difficult for readers without a strong background in mathematical modeling to follow. The choice of a recurrent neural network (RNN) for image segmentation tasks is also questionable, as convolutional neural networks (CNNs) are more commonly and effectively used for spatial data. Providing a rationale or comparative analysis for this choice would strengthen the methodology. Another limitation is the narrow scope of the dataset, which lacks diversity in terms of staining protocols and image types. Although augmentation is used, it does not fully guarantee the generalizability of the model. Additionally, the absence of publicly shared code or implementation details reduces the reproducibility of the study, which is a key aspect in academic research.
Authors’ Response:
- The grammatical errors and language have been sorted out and proofread. The mathematical content particularly the geometric active contour method equation has been inculcated into our paper thus the non-scientific reader has been accorded the option of accessing it from the cited authors.
- The choice of the RNN model is because of its sequential form of dealing with the various layers of the model handling the image data. Deep learning offers a CNN equivalent via the Tensor flow library and Keras architecture via the sequential model (RNN), LSTM, UNET among others.
- Thanks for the insight regarding rationale for comparative analysis of our method. That is a potential point of consideration in our future work.
- The scarcity of datasets is due to ethical concerns thus, in our case we had to resort to augmentation as our next cause of resolve. In our study, focus was on H&E images and the dataset we got from Kaggle repository has that as the stained protocol. The objective was to separate nuclei objects from tissue structures which our method clearly achieved.
- This has been addressed and included in the data availability section. This is highlighted in Blue in the paper.
Concern 3
Looking ahead, the paper points to several promising directions for future work, which align with the suggestions that emerge from the review. The authors could expand their testing to include other publicly available datasets, such as MoNuSeg, to validate the robustness of their approach. There is also potential to apply the method to histological images from other organs or with different staining techniques. Exploring modern machine learning architectures such as transformer-based models or attention mechanisms could further improve segmentation accuracy. Lastly, incorporating features like model explainability or confidence estimation could increase the clinical trust and usability of the segmentation results.
Authors’ Response:
- The practicability of our approach has been tested on the publicly available Warwick QU dataset from the Kaggle dataset repository. This dataset consists of other human glands histology images namely; the liver, colon, and kidney among others thus, it is applicable to other organs histology images though still dataset is limited in size. These concerns have been mentioned in the results and discussion section, and the future work section which is highlighted in Blue in the paper.
- Potential areas of interest has also been included to complement the existing ones in our study. These are highlighted in Blue in the paper. The reviewer suggestions will be added as part of the details of future work.

Round 2
Reviewer 2 Report (New Reviewer)
Comments and Suggestions for Authors
I find the paper suitable for publication in its present form.
This manuscript is a resubmission of an earlier submission. The following is a list of the peer review reports and author responses from that submission.
Round 1
Reviewer 1 Report
Comments and Suggestions for Authors
The manuscript demonstrates a 42% similarity index with the paper Automatic Watershed Segmentation of Cancerous Lesions in Unsupervised Breast Histology Images, authored by the lead author of this submission. This significant overlap indicates self-plagiarism. The current work heavily replicates the methodology, results, and discussions without sufficient innovation or differentiation from the prior publication. Although referencing one’s earlier work is acceptable, it should provide a foundation for novel contributions rather than duplicating content verbatim.
Methodology Limitations
-
Dependency on Image Preprocessing: The segmentation method relies extensively on data augmentation and stain normalization techniques. While these enhance dataset size and uniformity, they also introduce potential biases. For instance, the quality of the stain normalization affects the nuclei isolation, making the method less robust to real-world histology variations.
-
Limited Dataset: The dataset comprises only 24 images before augmentation. Although the number was increased through transformations, the lack of diverse, real-world data may affect the generalizability of the results. The augmented dataset lacks the complexity of naturally occurring variations in histopathology images.
-
Failure in Complex Overlaps: While the hybrid method addresses overlapping nuclei, its reliance on connected component analysis and active contours segmentation may struggle with images containing complex or dense overlaps. This could lead to over-segmentation or incorrect boundary detection.
-
Lack of External Validation: The method was validated solely on an augmented dataset. External validation on independent datasets or real-world images is crucial to assess the reliability and applicability of the proposed segmentation technique.
-
Limited Scalability to Other Modalities: The proposed method is tailored to specific breast histology images stained with H&E. It does not discuss the potential application or adaptation to other staining techniques or image modalities.
-
High False Positives in CAD Systems: The active contour method, while effective in segmentation, often produces false positives. The lack of a robust post-processing pipeline to address these errors further diminishes the reliability of the method for clinical adoption.
These limitations need to be addressed in the revised manuscript to ensure the novelty and practical applicability of the proposed segmentation approach.
Author Response
REVIEWER 1
Concern 1
The manuscript demonstrates a 42% similarity index with the paper Automatic Watershed Segmentation of Cancerous Lesions in Unsupervised Breast Histology Images, authored by the lead author of this submission. This significant overlap indicates self-plagiarism. The current work heavily replicates the methodology, results, and discussions without sufficient innovation or differentiation from the prior publication. Although referencing one’s earlier work is acceptable, it should provide a foundation for novel contributions rather than duplicating content verbatim.
Authors’ Response:
The similarity stems from where the watershed paper's future work section suggested other hybrid contributions would stem from. This clearly also highlighted in blue in the paper in the conclusion section on how exactly the two papers are not the same.
Concern 2
Methodology Limitations
- Dependency on Image Preprocessing: The segmentation method relies extensively on data augmentation and stain normalization techniques. While these enhance dataset size and uniformity, they also introduce potential biases. For instance, the quality of the stain normalization affects the nuclei isolation, making the method less robust to real-world histology variations.
Authors’ Response:
Image preprocessing is what mainly guides the whole segmentation step since active contours tend to introduce an increased number false-positives and false-negatives. Image preprocessing tends to reduce this number, you can either focus or rather introduce post-processing if you don’t realise significant results.
Concern 3
- Limited Dataset: The dataset comprises only 24 images before augmentation. Although the number was increased through transformations, the lack of diverse, real-world data may affect the generalizability of the results. The augmented dataset lacks the complexity of naturally occurring variations in histopathology images.
Authors’ Response:
Limited public available medical imaging datasets is out of reach due to ethical concerns, thus we work with what we can gain access to, and also data augmentation methods help to increase those dataset sizes.
Concern 4
- Failure in Complex Overlaps: While the hybrid method addresses overlapping nuclei, its reliance on connected component analysis and active contours segmentation may struggle with images containing complex or dense overlaps. This could lead to over-segmentation or incorrect boundary detection.
Authors’ Response:
This phenomenon has been explained quite clearly in the introduction and proposed method section, where we have meticulously talked about the essence of preprocessing and focused majorly on stain normalization. The essence of stain normalization in our method is to just to separate the nuclei objects from tissue regions, then active contours and connected components will deal with their individual tasks respectively as specified in the paper. The image pre-processing step plays a major role in the segmentation method in our paper.
Concern 5
- Lack of External Validation: The method was validated solely on an augmented dataset. External validation on independent datasets or real-world images is crucial to assess the reliability and applicability of the proposed segmentation technique.
Authors’ Response:
This has been addressed and is highlighted in blue in the paper.
Concern 6
- Limited Scalability to Other Modalities: The proposed method is tailored to specific breast histology images stained with H&E. It does not discuss the potential application or adaptation to other staining techniques or image modalities.
Authors’ Response:
This has been addressed and is highlighted in blue in the paper.
Concern 7
- High False Positives in CAD Systems: The active contour method, while effective in segmentation, often produces false positives. The lack of a robust post-processing pipeline to address these errors further diminishes the reliability of the method for clinical adoption.
Authors’ Response:
Image preprocessing is what mainly guides the whole segmentation step since active contours tend to introduce an increased number of false-positives and false negatives. Image preprocessing tends to reduce this number, you can either focus or rather introduce post-processing if you don’t realize significant results.

Reviewer 2 Report
Comments and Suggestions for Authors
Journal: Bioengineering
Title: Active Contours Connected Component Analysis segmentation method of cancerous lesions in unsupervised breast histology images
Manuscript ID: bioengineering-3450283
Authors: Vincent Majanga, Ernest Mnkandla, Zenghui Wang and Donatien Moulla,
Connected Component Analysis (CCA) is a well-known technique to simplify (binarize) color images, but CCA leads to information loss necessarily. I do not see any reason / advantage of this binarization.
Table 2 shows that the same data were studied abundantly. Moreover, at least two sources are highly competitive with the presented approach: Mahanta et al. [30] and Niaz et al. [18]. This fact does not favor publication. The authors should have carried out a statistical test about significance, i.e., whether the betterment is significant or not. Alone this table questions the significant novelty of this contribution.
In any case, significance cannot be determined by one test and especially by one performance parameter. The corresponding literature suggests the usage of at least six performance parameters: see e.g., Table 4 in ref. [ Rich Caruana and Alexandru Niculescu-Mizil, KDD-2004 - Proceedings of the Tenth ACM SIGKDD International Conference on Knowledge Discovery and Data Mining2004, pp. 69-78.; Seattle, WA; United States; 22-25 August 2004; Code 64210
https://doi.org/10.1145/1014052.1014063 ] or Table 6 [ Selahattin Barış Çelebi* and Bülent Gürsel Emiroğlu, A Novel Deep Dense Block-Based Model for Detecting Alzheimer’s Disease, Applied Sciences. 13 (2023) 8686. https://doi.org/10.3390/app13158686 ].
Naturally, the performance parameters behave contradictory. (They are conflicting.) Hence, one cannot improve one without worsening the other(s). Therefore, multicriteria decision analysis (MCDA i.e., Pareto optimization) is the only suitable choice.
One more deficiency hinders acceptance greatly. Namely, the validation aspects are totally missing. A dual split is not sufficient. A slit of training, validation and test sets were appropriate. Cross-validation, bootstrap, randomization test, a rotation by 90° of images—all are necessary to prove that the findings are valid and not by chance.
On the other hand, the language is perfect and easily understandable.
Minor errors
Figure 1 “performance evaluation metrics”: i) strictly speaking the common way of performance evaluation is not metrics, it can be called performance measures, indicators or merits, by no means metrics; ii) only one indicator (accuracy) has been used.
Misleading abbreviations and notations: e.g. “R^2 denote a family of curves” but R^2 has been used as determination coefficient widely and unambiguously, especially in the context of performance measures; “function ⋃,” The U has been applied as logical (Boolean) operator widely and unambiguously earlier. Nabla is a mathematical operator represented by the nabla symbol [ https://en.wikipedia.org/wiki/Del ]. It is bound to Cartesian coordinates see the definition part in the previous link.
“Fig. 8 illustrates the proposed method’s training and validation loss/accuracy graph”: The graph does not say much except that no more than seven epochs are needed, overfit appears after that.
Abbreviations should not be used in titles abstract and conclusion.
Conclusion
The first paragraph has nothing to do with the conclusion of the present work. Its place is in the Introduction (eventually in the abstract or summary).
“There is a huge potential use of CAD systems”: the authors are apparently do not know the “English Understatement” [ https://www.orchidenglish.com/british-understatement/ ]. Namely, “huge potential” means that it cannot be used so well as expected.
“This study discusses”: such fragments are superfluous and should be avoided. What else?
Etc. Etc.
The references are mainly from the Asian Scientific Circle, which cannot be supported.
In summary, this work embodies a rather this scientific material, the novelty is dubious to say the least. I was thinking a lot about how to transfer this manuscript into an acceptable one.
A correction of minor errors is far from being sufficient. More calculations, more modeling, more performance parameters and their multicriteria decision analysis are absolute necessary. Similarly, the validation aspects should be exhausted even the applicability domain should be given.
At present the manuscript cannot be accepted in its present form.
Feb 06 / 2025 referee:
Author Response
REVIEWER 2
Concern 1
Connected Component Analysis (CCA) is a well-known technique to simplify (binarize) color images, but CCA leads to information loss necessarily. I do not see any reason / advantage of this binarization.
Authors’ Response:
Image pre-processing is the most important step in our study to highlight and separate nuclei objects from tissue objects, after that CCA is just to connect those objects with similar properties thus no information loss.
Concern 2
Table 2 shows that the same data were studied abundantly. Moreover, at least two sources are highly competitive with the presented approach: Mahanta et al. [30] and Niaz et al. [18]. This fact does not favor publication. The authors should have carried out a statistical test about significance, i.e., whether the betterment is significant or not. Alone this table questions the significant novelty of this contribution.
Authors Response:
The significance of our study has been substantiated by tests in the conclusion section and the raised concerns have been highlighted in blue in the paper.
Concern 3
In any case, significance cannot be determined by one test and especially by one performance parameter. The corresponding literature suggests the usage of at least six performance parameters: see e.g., Table 4 in ref. [ Rich Caruana and Alexandru Niculescu-Mizil, KDD-2004 - Proceedings of the Tenth ACM SIGKDD International Conference on Knowledge Discovery and Data Mining2004, pp. 69-78.; Seattle, WA; United States; 22-25 August 2004; Code 64210
https://doi.org/10.1145/1014052.1014063 ] or Table 6 [ Selahattin Barış Çelebi* and Bülent Gürsel Emiroğlu, A Novel Deep Dense Block-Based Model for Detecting Alzheimer’s Disease, Applied Sciences. 13 (2023) 8686. https://doi.org/10.3390/app13158686 ].
Naturally, the performance parameters behave contradictory. (They are conflicting.) Hence, one cannot improve one without worsening the other(s). Therefore, multicriteria decision analysis (MCDA i.e., Pareto optimization) is the only suitable choice.
Authors Responses:
The choice of performance evaluation metric was the F1-accuracy score, and has proven to achieve significant results even with other human glands as highlighted in blue in the results section and conclusion section of this paper.
Concern 4
Figure 1 “performance evaluation metrics”: i) strictly speaking the common way of performance evaluation is not metrics, it can be called performance measures, indicators or merits, by no means metrics; ii) only one indicator (accuracy) has been used.
Authors Responses:
The raised concern has been addressed and is highlighted in blue in paper.
Concern 5
Misleading abbreviations and notations: e.g. “R^2 denote a family of curves” but R^2 has been used as determination coefficient widely and unambiguously, especially in the context of performance measures; “function ⋃,” The U has been applied as logical (Boolean) operator widely and unambiguously earlier. Nabla is a mathematical operator represented by the nabla symbol [ https://en.wikipedia.org/wiki/Del ]. It is bound to Cartesian coordinates see the definition part in the previous link.
Authors Responses:
The raised concern has been addressed and is highlighted in blue in paper.
Concern 6
“Fig. 8 illustrates the proposed method’s training and validation loss/accuracy graph”: The graph does not say much except that no more than seven epochs are needed, overfit appears after that.
Authors Responses:
Several tests were iteratively undertaken for 75, 60, 50,40, 30, 20 epochs, and 30 epochs was deemed optimal and thus preferred. 20 epochs and below was underfitting thus could be considered.
Concern 7
Abbreviations should not be used in titles abstract and conclusion.
Conclusion
The first paragraph has nothing to do with the conclusion of the present work. Its place is in the Introduction (eventually in the abstract or summary).
“There is a huge potential use of CAD systems”: the authors are apparently do not know the “English Understatement” [ https://www.orchidenglish.com/british-understatement/ ]. Namely, “huge potential” means that it cannot be used so well as expected.
“This study discusses”: such fragments are superfluous and should be avoided. What else?
Etc. Etc.
The references are mainly from the Asian Scientific Circle, which cannot be supported.
Authors Responses:
All raised concerns have been addressed and are highlighted in blue in the paper.

Reviewer 3 Report
Comments and Suggestions for Authors
-
Major issues
- The proposed ACCCA (Active Contours Connected Component Analysis) method reports a high F1-score of 98.71%. However, it is unclear whether the model's performance has been validated on an external dataset beyond the one used for training. To ensure robustness, has the method been tested on publicly available datasets to confirm its generalization ability? Additionally, does the manuscript provide sufficient implementation details, including hyperparameter settings, preprocessing steps, and source code, to ensure reproducibility?
- The study aims to address the limitations of conventional Active Contour methods by integrating Connected Component Analysis (CCA) to improve segmentation of overlapping nuclei. However, it is unclear how much performance improvement is achieved compared to previous works. A quantitative comparison table with competing segmentation techniques (e.g., accuracy, processing time, computational efficiency) is necessary to highlight the proposed method’s advantages. Has the study explicitly evaluated its segmentation performance against state-of-the-art Active Contour-based approaches to justify its novelty?
Minor issues
- The manuscript describes data augmentation (rotation, shifting, scaling) and stain normalization to enhance dataset diversity. However, it is unclear whether these transformations preserve clinically relevant features in histopathological images. Were any objective evaluations conducted to confirm the effectiveness of stain normalization in maintaining meaningful color features?
- The study employs an RNN-based segmentation model optimized using Adam with a specific learning rate, batch size, and dropout rate. However, the manuscript lacks a clear rationale for selecting these hyperparameters. Did the authors conduct hyperparameter tuning experiments (e.g., grid search, cross-validation), or were the parameters chosen arbitrarily?
- While the study suggests that CCA improves the performance of Active Contours, it does not provide a clear performance comparison between the methods. Has an ablation study been conducted to demonstrate how Active Contours alone, CCA alone, and their combination perform on segmentation tasks? Providing quantitative evidence of each method’s contribution would strengthen the study’s claims.
- The paper includes some visual examples of segmentation outputs, but side-by-side comparisons with existing methods are lacking. Could the authors provide comparative visualizations (e.g., displaying how the proposed method improves boundary accuracy compared to prior techniques)?
- The study primarily evaluates segmentation performance but does not explicitly discuss its practical applicability in a clinical setting. How does the proposed method enhance pathologists’ workflows, and has it been evaluated in a real-world diagnostic scenario? Are there any computational constraints that might limit its usability in real-time histopathological analysis?
Author Response
REVIEWER 3
Concern 1
- The proposed ACCCA (Active Contours Connected Component Analysis) method reports a high F1-score of 98.71%. However, it is unclear whether the model's performance has been validated on an external dataset beyond the one used for training. To ensure robustness, has the method been tested on publicly available datasets to confirm its generalization ability? Additionally, does the manuscript provide sufficient implementation details, including hyperparameter settings, preprocessing steps, and source code, to ensure reproducibility?
- The study aims to address the limitations of conventional Active Contour methods by integrating Connected Component Analysis (CCA) to improve segmentation of overlapping nuclei. However, it is unclear how much performance improvement is achieved compared to previous works. A quantitative comparison table with competing segmentation techniques (e.g., accuracy, processing time, computational efficiency) is necessary to highlight the proposed method’s advantages. Has the study explicitly evaluated its segmentation performance against state-of-the-art Active Contour-based approaches to justify its novelty?
Authors Responses:
The raised concerns have been addressed in the results and discussion section and are highlighted in blue in the paper.
Concern 2
The manuscript describes data augmentation (rotation, shifting, scaling) and stain normalization to enhance dataset diversity. However, it is unclear whether these transformations preserve clinically relevant features in histopathological images. Were any objective evaluations conducted to confirm the effectiveness of stain normalization in maintaining meaningful color features?
Authors Responses:
This has been explained quite clearly in the introduction and proposed method section, where we have meticulously talked about the essence of preprocessing and focused majorly on stain normalization. The essence of stain normalization in our method is just to highlight and separate the nuclei objects from tissue regions, then active contours and connected components will deal with their individual tasks respectively as specified in the paper. The image pre-processing step plays a major role in the segmentation method in our paper.
Concern 3
The study employs an RNN-based segmentation model optimized using Adam with a specific learning rate, batch size, and dropout rate. However, the manuscript lacks a clear rationale for selecting these hyperparameters. Did the authors conduct hyperparameter tuning experiments (e.g., grid search, cross-validation), or were the parameters chosen arbitrarily?
Authors Responses:
The raised concerns have been addressed in the results and discussion section and are highlighted in blue in the paper.
Concern 4
While the study suggests that CCA improves the performance of Active Contours, it does not provide a clear performance comparison between the methods. Has an ablation study been conducted to demonstrate how Active Contours alone, CCA alone, and their combination perform on segmentation tasks? Providing quantitative evidence of each method’s contribution would strengthen the study’s claims.
Authors Responses:
Table 2 compares those methods related to breast cancer segmentation and are linked to the application of active contours, CCA methods were also highlighted but majorly those linked with active contours.
Concern 5
The paper includes some visual examples of segmentation outputs, but side-by-side comparisons with existing methods are lacking. Could the authors provide comparative visualizations (e.g., displaying how the proposed method improves boundary accuracy compared to prior techniques)?
The study primarily evaluates segmentation performance but does not explicitly discuss its practical applicability in a clinical setting. How does the proposed method enhance pathologists’ workflows, and has it been evaluated in a real-world diagnostic scenario? Are there any computational constraints that might limit its usability in real-time histopathological analysis?
Authors Responses:
These raised concerns have been addressed meticulously in the results discussion, and conclusion section, and are highlighted in blue in the paper.

Reviewer 4 Report
Comments and Suggestions for Authors
Dear authors,
the related work should be in the introduction section. Based on the description of the related work you should summarize the related work in the table providing reference number, methods they have used and the results they have obtained. Based on the related work you should write what are dsiadvantages of the the related work and just after that you should write the novelty and the idea of your work. Why is your work better than the previous work.
Conclusions should contain the answers to the hypotheses defined in the introduction section and structured in the bullet format.
Author Response
REVIEWER 4
Concern 1
the related work should be in the introduction section. Based on the description of the related work you should summarize the related work in the table providing reference number, methods they have used and the results they have obtained. Based on the related work you should write what are dsiadvantages of the the related work and just after that you should write the novelty and the idea of your work. Why is your work better than the previous work
Authors’ Response:
The raised concerns have been addressed and are highlighted in blue in the paper.
Concern 2
Conclusions should contain the answers to the hypotheses defined in the introduction section and structured in the bullet format.
Authors’ Response:
The raised concern has been addressed and is highlighted in blue in the paper
Authors Responses:
The raised concern has been addressed and is highlighted in blue in paper.

Round 2
Reviewer 2 Report
Comments and Suggestions for Authors
Journal: Bioengineering
Title: Active Contours Connected Component Analysis segmentation method of cancerous lesions in unsupervised breast histology images
Manuscript ID: bioengineering-3450283V2
Authors: Vincent Majanga, Ernest Mnkandla, Zenghui Wang and Donatien Moulla,
Authors response
Concern 1
I do not see any reason / advantage of this binarization.
Authors’ Response:
Image pre-processing is the most important step in our study to highlight and separate nuclei objects from tissue objects, after that CCA is just to connect those objects with similar properties thus no information loss.
The authors’ answer is far from satisfactory. You should not be a scientist to see that a color image is much more informative than a black and white one. In any case, an important step in a bad technique is not acceptable.
Concern 2
Table 2 shows that the same data were studied abundantly. Moreover, at least two sources are highly competitive with the presented approach: Mahanta et al. [30] and Niaz et al. [18]. This fact does not favor publication.
The Authors’ Response is missing.
Concern 2 (cont.)
The authors should have carried out a statistical test about significance, i.e., whether the betterment is significant or not.
Authors Response:
The significance of our study has been substantiated by tests in the conclusion section and the raised concerns have been highlighted in blue in the paper.
The authors’ answer is far from satisfactory. No tests have been made, and no test is mentioned in the conclusion part. No blue highlights can be seen in my version.
Concern 2 (cont.)
Alone this table [Table 2] questions the significant novelty of this contribution.
The authors were oblivious to answer this issue. I wonder how they can prove significant novelty when the betterment is negligible, if at all.
Concern 3
In any case, significance cannot be determined by one test and especially by one performance parameter.
NO answer from the authors.
Concern 3 (cont.)
The corresponding literature suggests the usage of at least six performance parameters: see e.g., Table 4 in ref. [ Rich Caruana and Alexandru Niculescu-Mizil, KDD-2004 - Proceedings of the Tenth ACM SIGKDD International Conference on Knowledge Discovery and Data Mining2004, pp. 69-78.; Seattle, WA; United States; 22-25 August 2004; Code 64210
https://doi.org/10.1145/1014052.1014063 ] or Table 6 [ Selahattin Barış Çelebi* and Bülent Gürsel Emiroğlu, A Novel Deep Dense Block-Based Model for Detecting Alzheimer’s Disease, Applied Sciences. 13 (2023) 8686. https://doi.org/10.3390/app13158686 ].
The authors have not made any effort to read the suggested papers, learn the proper validation/testing practices, and have not acted accordingly.
Concern 3 (cont.)
Naturally, the performance parameters behave contradictory. (They are conflicting.) Hence, one cannot improve one without worsening the other(s). Therefore, multicriteria decision analysis (MCDA i.e., Pareto optimization) is the only suitable choice.
Authors Responses:
The choice of performance evaluation metric was the F1-accuracy score, and has proven to achieve significant results even with other human glands as highlighted in blue in the results section and conclusion section of this paper.
The authors’ answer is far from satisfactory. If one select one performance parameters only, the contradictory character cannot be seen. The authors have not even realized the conflict, the need of more evaluation indicators [NOT metric for heaven’s sake!], and the necessity of multicriteria decision making. As said, no test has been completed for statistical significance. No blue highlights can be seen in my version.
Concern 4
Figure 1 “performance evaluation metrics”: i) strictly speaking the common way of performance evaluation is not metrics, it can be called performance measures, indicators or merits, by no means metrics; ii) only one indicator (accuracy) has been used.
Authors Responses:
The raised concern has been addressed and is highlighted in blue in paper.
The authors repeatedly use the wrong term metrics. The earlier answer is NOT acceptable.
Concern 5
Misleading abbreviations and notations: e.g. “R^2 denote a family of curves” but R^2 has been used as determination coefficient widely and unambiguously, especially in the context of performance measures; “function ⋃,” The U has been applied as logical (Boolean) operator widely and unambiguously earlier. Nabla is a mathematical operator represented by the nabla symbol [ https://en.wikipedia.org/wiki/Del ]. It is bound to Cartesian coordinates see the definition part in the previous link.
Authors Responses:
The raised concern has been addressed and is highlighted in blue in paper.
The R^2, nabla and “function ⋃,” remained in the manuscript.
Concern 6
“Fig. 8 illustrates the proposed method’s training and validation loss/accuracy graph”: The graph does not say much except that no more than seven epochs are needed, overfit appears after that.
Authors Responses:
Several tests were iteratively undertaken for 75, 60, 50,40, 30, 20 epochs, and 30 epochs was deemed optimal and thus preferred. 20 epochs and below was underfitting thus could be considered.
It is sufficient to have a superficial look in Figure 8 that overfit appears after ~seven epoch. The authors do not even understand the terms they are using.
Concern 7
Abbreviations … Asian Scientific Circle, which cannot be supported.
Authors’ Responses:
All raised concerns have been addressed and are highlighted in blue in the paper.
Correction of all minor errors does not help to accept the manuscript.
The usage of artificial intelligence (AI) is forbidden in the scientific practice at present. No serious publisher (Wiley, ACS, Elsevier, and MDPI) allows writing results, discussion and conclusion parts by AI, except language corrections. The AI application is indicated by the sectioning of the article, especially the future work.
The manuscript should be rejected on ethical grounds without the option of resubmission.
Author Response
REVIEWER 2
Concern 1
I do not see any reason / advantage of this binarization
Authors Responses:
The essence of the whole binarization process is to highlight exact positions of the cancerous lesions on the image. We have provided both image outputs for clarity purposes.
Concern 2
Table 2 shows that the same data were studied abundantly. Moreover, at least two sources are highly competitive with the presented approach: Mahanta et al. [30] and Niaz et al. [18]. This fact does not favor publication.
Authors Responses:
The results are highly competitive but not superior thus preference of the proposed method.
Concern 3
The authors’ answer is far from satisfactory. No tests have been made, and no test is mentioned in the conclusion part. No blue highlights can be seen in my version
Authors Responses:
Testing via the practicability of the proposed method has been provided in results and discussion section, where the method has been tested even with other different images from the QU warwick dataset together with the accuracy score obtained on it.
Concern 4
Alone this table [Table 2] questions the significant novelty of this contribution.
The authors were oblivious to answer this issue. I wonder how they can prove significant novelty when the betterment is negligible, if at all.
.
Authors Responses:
Contribution has been made by the proposed method irrespective of how negligible.
Concern 5
The authors have not made any effort to read the suggested papers, learn the proper validation/testing practices, and have not acted accordingly.
Authors Responses:
Testing practices are not standard; we chose the practicability angle and it seems to provide significant results based on the accuracy scores got.
Concern 6
The authors’ answer is far from satisfactory. If one select one performance parameters only, the contradictory character cannot be seen. The authors have not even realized the conflict, the need of more evaluation indicators [NOT metric for heaven’s sake!], and the necessity of multicriteria decision making. As said, no test has been completed for statistical significance. No blue highlights can be seen in my version.
Authors Responses:
Testing practices are not standard; we chose the practicability angle and it seems to provide significant results based on the accuracy scores got.
Concern 7
The R^2, nabla and “function ⋃,” remained in the manuscript.
Authors Responses:
Refer to the referenced papers 42 and 43 for clear understanding of the latex symbols.
Concern 7
It is sufficient to have a superficial look in Figure 8 that overfit appears after ~seven epoch. The authors do not even understand the terms they are using.
Authors Responses:
Overfitting is understood by the researchers, and is observed on the validation accuracy and the training accuracy, and on the entire graph which includes all 30 epochs not just the seventh epoch. In the different epochs 75, 50, and 40 that were observed after early stopping, the shared result was the best.

Reviewer 3 Report
Comments and Suggestions for Authors
My concerns have been satisfactorily addressed, and I believe the work is now ready for publication.
Comments on the Quality of English LanguageMy concerns have been satisfactorily addressed, and I believe the work is now ready for publication.
Author Response
Thanks for your kind response.
Reviewer 4 Report
Comments and Suggestions for Authors
Dear authors,
you have corrected the manuscript according to my comments and suggestions. The manuscript can be accepted in this form.
Author Response
Thanks.